# Template-free Articulated Gaussian Splatting for Real-time Reposable Dynamic View Synthesis

**Diwen Wan**[1]    **Yuxiang Wang**[1]    **Ruijie Lu**[1]    **Gang Zeng**[1]

[1]National Key Laboratory of General Artificial Intelligence,
School of IST, Peking University, China
{wan,yuxiang123}@stu.pku.edu.cn   {jason_lu,zeng}@pku.edu.cn

## Abstract

While novel view synthesis for dynamic scenes has made significant progress, capturing skeleton models of objects and re-posing them remains a challenging task. To tackle this problem, in this paper, we propose a novel approach to automatically discover the associated skeleton model for dynamic objects from videos without the need for object-specific templates. Our approach utilizes 3D Gaussian Splatting and superpoints to reconstruct dynamic objects. Treating superpoints as rigid parts, we can discover the underlying skeleton model through intuitive cues and optimize it using the kinematic model. Besides, an adaptive control strategy is applied to avoid the emergence of redundant superpoints. Extensive experiments demonstrate the effectiveness and efficiency of our method in obtaining re-posable 3D objects. Not only can our approach achieve excellent visual fidelity, but it also allows for the real-time rendering of high-resolution images. Please visit our project page for more results: `https://dnvtmf.github.io/SK_GS/`.

## 1 Introduction

Novel view synthesis for 3D scenes is important for many domains including virtual/augmented/mixed reality, game or movie productions. In recent years, Neural Radiance Fields (NeRF) [1] have witnessed significant advances in both static and dynamic scenes. Among them, 3D Gaussian Splatting (3D-GS) [2] proposed a novel point-based representation, and is capable of real-time rendering while ensuring the quality of generated images, bringing new insights to more complex task scenarios.

Although visually compelling results and fast rendering speed have been achieved in reconstructing a dynamic scene, current methods mainly focus on replaying the motion in the video, which means it just renders novel view images within the given time range, making it hard to explicitly repose or control the movement of individual objects in the scene. For some specific categories such as the human body or human head, one main approach is to leverage the category-specific prior knowledge such as templates to support the manipulation of reconstructed objects. However, it is hard for these methods to generalize to large-scale in-the-wild scenes or human-made articulated objects.

Some template-free methods attempt to address these challenges by building reposable models from videos. Watch-It-Move (WIM) [3] leverages ellipsoids, an explicit representation, to coarsely model 3D objects, and then estimate the residual by a neural network. The underlying intuition is that one or more ellipsoids can represent a functional part. By observing the motion of parts from multi-view videos, WIM can learn both the appearance and structure of articulated objects. However, the reconstruction results of WIM are of low visual quality and the training and rendering speed is slow. Apart from WIM, Articulated Point NeRF (AP-NeRF) [4] samples feature point cloud from a pre-trained dynamic NeRF model (TiNeuVox [5]) and initializes the skeleton tree using the medial axis transform algorithm. By combining linear blend skinning (LBS) and point-based rendering [6],

38th Conference on Neural Information Processing Systems (NeurIPS 2024).

AP-NeRF jointly optimizes dynamic NeRF and skeletal model from videos. Compared to WIM, AP-NeRF achieves higher visual fidelity while significantly reducing the training time. However, AP-NeRF cannot achieve real-time rendering, which is still far from practical application.

In this paper, we target class-agnostic novel view synthesis of reposable models without the need for a template or pose annotations, while achieving real-time rendering. To enable fast rendering speed, we opt to represent the 3D object as 3D Gaussian Splatting. To be specific, we first reconstruct the 3D dynamic model using 3D Gaussians and superpoints, where each superpoint binds Gaussians with similar motions together. These superpoints will later be treated as the parts of an object. Afterward, a skeleton model is discovered leveraging some intuitive cues under the guidance of superpoint motions from the video. Finally, we jointly optimize the skeleton model and pose parameters to match the motions of the training videos. During the optimization process of object reconstruction, we will inevitably generate a lot of redundant superpoints to fit the complex motion. To simplify the skeleton model and avoid overfitting, we employ an adaptive control strategy and regularization losses to reduce the number of superpoints. Our contributions can be summarized as follows:

- We propose a novel method based on 3D Gaussians and superpoints for modeling appearance, skeleton model, and motion of articulated dynamic objects from videos. Our approach can automatically discover the skeleton model without any category-specific prior knowledge.
- We effectively learn and control superpoints by employing an adaptive control strategy and regularization losses.
- We demonstrate excellent novel view synthesis quality while achieving real-time rendering on various datasets.

## 2 Related Works

### 2.1 Static and Dynamic Neural Radiance Fields

In recent years, we have witnessed significant progress in the field of novel view synthesis empowered by Neural Radiance Fields. While vanilla NeRF [1] manages to synthesize photorealistic images for any viewpoint using MLPs, subsequent works have explored various representations such as 4D tensors [7], hash encodings [8], or other well-designed data structures [9, 10] to improve rendering quality and speed. More recently, a novel framework 3D Gaussian Splatting [2] has received widespread attention for its ability to synthesize high-fidelity images for complex scenes in real-time.

Meanwhile, many research works challenge the hypothesis of a static scene in NeRFs and attempt to synthesize novel-view images of a dynamic scene at an arbitrary time from a 2D video, which is a more challenging task since the correspondence between different frames is non-trivial. One line of research works [11–13] directly represents the dynamic scene with an additional time dimension or a time-dependent interpolation in a latent space. Another line of work [14–18] represents the dynamic scene as a static canonical 3D scene along with its deformation fields. While one main bottleneck of synthesizing a dynamic scene is speed, some works [19–22] propose to extend 3D Gaussian Splatting into 4D to mitigate the problem. Though being able to recover a high-fidelity scene, this method cannot directly support editing and reposing objects within it. In this work, we leverage 3D Gaussian Splatting as the representation for faster rendering speed.

### 2.2 Object Reposing

It's impractical to directly repose the deformation fields of dynamic NeRFs due to the complexity of high-dimension. Therefore, utilizing parametric templates based on object priors to represent deformation is adopted in many research works. The classes of parametric templates range from human faces [23, 24], and bodies [25, 26] to non-human objects like animals [27]. With the help of skeleton-based LBS and 3D or 2D annotations, these parametric templates are capable of representing articulate human heads [28–33] and bodies [34–41]. Though these template-based reposing methods can synthesize high-fidelity images, they are restricted to certain object classes and mainly deal with rigid motions, not to mention the time-consuming process of annotations.

To alleviate the excessive reliance on domain-specific skeletal models, methods based on retrieval from database [42] or adaptation from a generic graph [43, 44] are adopted. However, these methods are still of relatively low flexibility and diversity. Another line of work attempts to learn a more

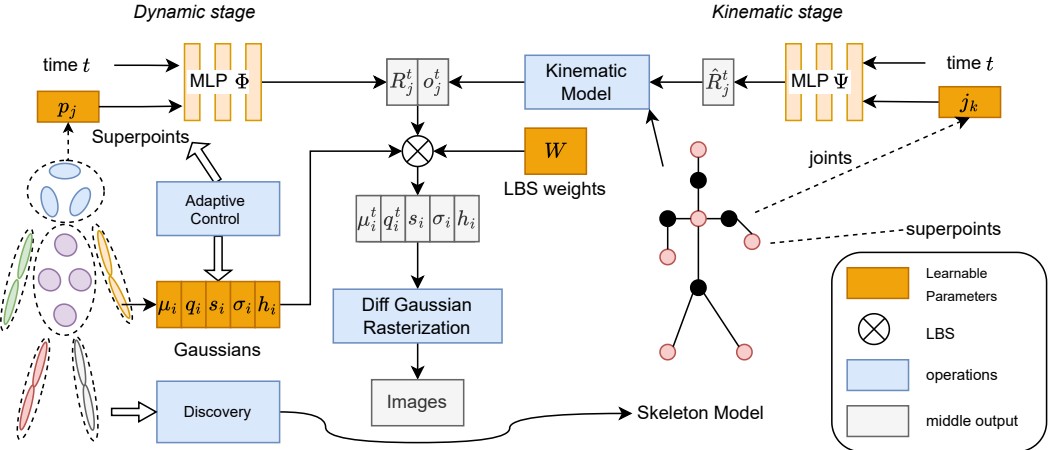

Figure 1: The pipeline of proposed approach. Our approach follows a two-stage training strategy. In the first stage (*i.e. dynamic* stage), we learn the 3D Gaussians and superpoints to reconstruct the appearance. Each superpoint is associated with a rigid part, and the adaptive control strategy is used to control the count. After finishing the training of *dynamic* stage, we can discover the skeleton model based on superpoints. After we finish the second stage (*i.e.*, *kinematic* stage), we can obtain an articulated model based on the kinematic model.

general template-free object representation by 3D shape recovery [3, 4, 45]. WIM [3] proposes to jointly learn a surface representation and LBS model for articulation without any supervision or prior knowledge of the structure. However, the reposing images are of low visual quality and the required training time is considerably long. AP-NeRF [4] achieves a much faster training speed by leveraging a point-based NeRF representation, but cannot support real-time rendering as well.

## 3 Methods

Our goal is to reconstruct a reposable articulated object with real-time rendering speed from videos. The pipeline of proposed method is illustrated in Fig. 1. We represent the appearance of the articulated object as 3D Gaussians in the canonical space while aggregating 3D Gaussians with similar motion into superpoints, which can be treated as rigid parts. It is noteworthy that we apply a time-variant 6 DoF transformation matrix to model the motion of the object. Based on these superpoints, we leverage several intuitive observations to guide the discovery of the skeleton model, which includes both joints and skeletons. Since the observations hold for most objects, our method does not require a category-specific template or pose annotations. To reduce the redundant superpoint, we propose an adaptive control strategy to densify, prune, and merge superpoints during the training process.

### 3.1 Preliminaries: 3D Gaussian Splatting

3D Gaussian Splatting (3D-GS) [2] use a set of 3D Gaussians to represent a 3D scene. Each Gaussian $G_i$: $\{\boldsymbol{\mu}_i, \boldsymbol{q}_i, \boldsymbol{s}_i, \sigma_i, \boldsymbol{h}_i\}$ is associated with a position $\boldsymbol{\mu}_i$, a rotation matrix $\mathbf{R}_i$ which is parameterized by a quaternion $\boldsymbol{q}_i$, a scaling matrix $\mathbf{S}_i$ which is parameterized by a 3D vector $\boldsymbol{s}_i$, opacity $\sigma_i$ and spherical harmonics (SH) coefficients $\boldsymbol{h}_i$. Therefore, the anisotropic 3D covariance matrix of Gaussian $G_i$ is defined as $\boldsymbol{\Sigma}_i = \mathbf{R}_i \mathbf{S}_i \mathbf{S}_i^\top \mathbf{R}_i^\top$, which is positive semi-definite matrix.

To render images, 3D-GS employs EWA Splatting algorithm [46] to project a 3D Gaussian with center $\boldsymbol{\mu}_i$ and covariance $\boldsymbol{\Sigma}_i$ to 2D image space, and the projection can be approximated as a 2D Gaussian with center $\boldsymbol{\mu}_i'$ and covariance $\boldsymbol{\Sigma}_i'$. Let $\mathbf{Q}, \mathbf{K}$ be the viewing transformation and projection matrix, $\boldsymbol{\mu}_i'$ and $\boldsymbol{\Sigma}_i'$ are computed as

$$\boldsymbol{\mu}_i' = \mathbf{K}(\mathbf{Q}\boldsymbol{\mu}_i)/(\mathbf{Q}\boldsymbol{\mu}_i)_z, \quad \boldsymbol{\Sigma}_i' = \mathbf{J}\mathbf{Q}\boldsymbol{\Sigma}_i\mathbf{Q}^\top\mathbf{J}^\top, \tag{1}$$

where $\mathbf{J}$ is the Jacobian of the projective transformation. Therefore, the final opacity of a 3D Gaussian at pixel coordinate $\boldsymbol{x}$ is

$$\alpha_i = \sigma_i \exp(-\frac{1}{2}(\boldsymbol{x} - \boldsymbol{\mu}_i')^\top \boldsymbol{\Sigma}_i'^{-1}(\boldsymbol{x} - \boldsymbol{\mu}_i')) \tag{2}$$

After sorting Gaussians by depth, the color at $\boldsymbol{x}$ can be computed by volume rendering:

$$I = \sum_{i=1}^{N}(\boldsymbol{c}_i\alpha_i \prod_{j=1}^{i-1}(1 - \alpha_j)) \tag{3}$$

where RGB color $\boldsymbol{c}_i$ is evaluated by SH with coefficients $\boldsymbol{h}_i$ and view direction.

Given multi-view images with known camera poses, 3D-GS optimizes a static 3D scene by minimizing the following loss function:

$$\mathcal{L}_{rgb} = (1 - \lambda)\mathcal{L}_1(I, I_{gt}) + \lambda\mathcal{L}_{\text{SSIM}}(I, I_{gt}) \tag{4}$$

where $\lambda = 0.2$, and $I_{gt}$ is the ground truth. Besides, 3D-GS is initialized from from random point cloud or SfM sparse point cloud, and an adaptive density adjustment strategy is applied to control the number of Gaussians.

## 3.2 Dynamic Stage

To reconstruct an articulated object, we build the deformation based on superpoints and the linear blend skinning (LBS) [47], while the canonical model is represented by 3D-GS.

The superpoints $\mathcal{P} = \{\boldsymbol{p}_j \in \mathbb{R}^3\}_{j=1}^M$ are associated with a set of 3D Gaussians, and can be used to represent the object's rigid parts. For timestamp $t$, we directly use deformable field $\Phi$ to learn time-variant 6 DoF transformation $[\mathbf{R}_j^t, \boldsymbol{o}_j^t] \in \mathbf{SE(3)}$ of superpoint $\boldsymbol{p}_j$ as:

$$\Phi : (\boldsymbol{p}_j, t) \rightarrow (\mathbf{R}_j^t, \boldsymbol{o}_j^t), \tag{5}$$

where $\mathbf{R}_j^t \in \mathbf{SO(3)}$ is the local rotation matrix and $\boldsymbol{o}_j^t \in \mathbb{R}^3$ is the translation vector. Then, LBS is employed to derive the motion of each Gaussian by interpolating the transformations for their neighboring superpoints:

$$\boldsymbol{\mu}_i^t = \sum_{j \in \mathcal{N}_i} w_{ij}(\mathbf{R}_j^t\boldsymbol{\mu}_i + \boldsymbol{o}_j^t), \quad \boldsymbol{q}_i^t = (\sum_{j \in \mathcal{N}_i} w_{ij}\boldsymbol{r}_j^t) \otimes \boldsymbol{q}_i, \tag{6}$$

where $\boldsymbol{r}_j^t \in \mathbb{R}^4$ is the quaternion representation for matrix $\mathbf{R}_j^t$, and $\otimes$ is the production of quaternions. $\mathcal{N}_i$ denotes the $K$-nearest superpoints of Gaussian $G_i$. $w_{ij}$ is the LBS weights between Gaussian $G_i$ and superpoint $\boldsymbol{p}_j$, which can be computed as:

$$w_{ij} = \frac{\exp(\mathbf{W}_{ij})}{\sum_{k \in \mathcal{N}_i} \exp(\mathbf{W}_{ik})}, \tag{7}$$

where $\mathbf{W} \in \mathbb{R}^{N \times M}$ is a learnable parameter.

While keeping other attributes (*i.e.*, $\boldsymbol{s}_i, \sigma_i, \boldsymbol{h}_i$) of Gaussians the same as canonical space, we can render the image at timestamp $t$ following Eq. 3.

## 3.3 Discovery of Skeleton Model

Treating each superpoint as a rigid part of the articulated object, we can discover the skeleton model (*i.e.*, the 3D joints and the connection between joints) based on the motion of superpoints. Similar to WIM [3], there are some observations to help us discover the underlying skeleton. First, if there is a joint between two superpoints $\boldsymbol{p}_a$ and $\boldsymbol{p}_b$, the position of $\boldsymbol{p}_a$ is more likely close to the position of $\boldsymbol{p}_b$. Second, when the relative pose between two parts changes, the joint between the two parts is relatively unchanged. Lastly, two connected parts can be merged if they maintain the same relative pose throughout the whole sequence.

Let $\boldsymbol{j}_{ab} \in \mathbb{R}^3$ be the position of underlying joint between superpoints $\boldsymbol{p}_a$ and $\boldsymbol{p}_b$, and $\mathbf{R}_{ab}^t \in \mathbf{SO(3)}$ is the relative rotation matrix between two superpoints at time $t$. The relative transform between $\boldsymbol{p}_a$ and $\boldsymbol{p}_b$ can be either represented by the global transform or the rotation of the joint, that is:

$$\begin{bmatrix} \mathbf{R}_r^t & \boldsymbol{t}_r^t \\ \mathbf{0} & 1 \end{bmatrix} = \begin{bmatrix} \mathbf{R}_b^t & \boldsymbol{o}_b^t \\ \mathbf{0} & 1 \end{bmatrix}^{-1} \begin{bmatrix} \mathbf{R}_a^t & \boldsymbol{o}_a^t \\ \mathbf{0} & 1 \end{bmatrix} = \begin{bmatrix} \mathbf{R}_{ab}^t & \boldsymbol{j}_{ab} - \mathbf{R}_{ab}^t\boldsymbol{j}_{ab} \\ \mathbf{0} & 1 \end{bmatrix} \tag{8}$$

where $\mathbf{R}_r^t = (\mathbf{R}_b^t)^{-1}\mathbf{R}_a^t = \mathbf{R}_{ab}^t \in \mathbf{SO(3)}$ and $\boldsymbol{t}_r^t \in \mathbb{R}^3$ are the relative rotation matrix and translation vector between two superpoints respectively. Considering two joints $\boldsymbol{j}_{ab}$ and $\boldsymbol{j}_{ba}$ between $\boldsymbol{p}_a$ and $\boldsymbol{p}_b$ should be the same, we compute following distance $d_{ab}$ for every superpoint pair $(a, b)$:

$$d_{ab} = \sum_t \|\boldsymbol{t}_r - (\boldsymbol{j}_{ab} - \mathbf{R}_{ab}^t \boldsymbol{j}_{ab})\|_2^2 + \lambda_d \|\boldsymbol{j}_{ab} - \boldsymbol{j}_{ba}\|_2^2 \qquad (9)$$

where $\lambda_d = 1$ is the hyper-parameter. To prevent the distance from changing too quickly, we smooth it across training iterations:

$$\hat{d}_{ab}(\tau + 1) = (1 - \epsilon) \cdot \hat{d}_{ab}(\tau) + \epsilon \cdot d_{ab}(\tau), \qquad (10)$$

where $\epsilon = 0.1$ is the momentum, and $\tau$ is the training iteration.

Similar to WIM [3], we discover the structure $\Gamma$ of joints based on the distance $\hat{d}_{ab}$ by the minimum spanning tree algorithm. We first select all pairs $(a, b)$ if superpoint $\boldsymbol{p}_b$ is $K'$-nearest neighborhood for superpoint $\boldsymbol{p}_a$ and sort the list of $\hat{d}_{ab}$ for those pairs in ascending order. We initialize $\Gamma$ as an empty set. We pick pair $(a, b)$ from the lowest distance to the highest distance, and add this pair to $\Gamma$ while there is no path between $a$ and $b$. After finishing the procedure, we obtain the final object structure $\Gamma$, which is an acyclic graph, *i.e.*, a tree. We choose the node whose length of the longest path from itself to any other node is the shortest as the root node. If there is more than one candidate node, we randomly choose one as the root.

## 3.4 Kinematic Stage

After discovering the skeleton model, we optimize the skeleton model and fine-tune 3D Gaussians by using the kinematic model. Specifically, we first predict time-variant rotations $\hat{\mathbf{R}}_k \in \mathbf{SO(3)}$ for each joint $\boldsymbol{j}_k$ by using an deformable field $\Psi$:

$$\Psi : (\boldsymbol{j}_k, t) \to \hat{\mathbf{R}}_k \qquad (11)$$

Then we forward-warp the superpoint $\boldsymbol{p}_j$ from the canonical space to the observation space of timestamp $t$ via the kinematic model. The local transformation matrix $\hat{\mathbf{T}}_k^t \in \mathbf{SE(3)}$ of each joint $k$ is defined by a rotation $\mathbf{R}_k^t$ around its parent joint $\boldsymbol{j}_k$. Consequently, the final transformation of each superpoint $\boldsymbol{p}_j$ can be computed as a linear combination of bone transformation:

$$\mathbf{T}_j^t = \begin{bmatrix} \mathbf{R}_j^t & \boldsymbol{o}_j^t \\ \mathbf{0} & 1 \end{bmatrix} = \mathbf{T}_{root}^t \prod_{k \in \mathbb{C}_j} \hat{\mathbf{T}}_k^t, \text{ where } \hat{\mathbf{T}}_k^t = \begin{bmatrix} \hat{\mathbf{R}}_k^t & \boldsymbol{j}_k - \hat{\mathbf{R}}_k^t \boldsymbol{j}_k \\ \mathbf{0} & 1 \end{bmatrix}, \qquad (12)$$

where $\mathbb{C}_j$ is the list of ancestor of superpoint $j$ in skeleton model. $\mathbf{T}_{root}^t$ is global transformation of root. Same as Sec. 3.2, we use LBS to derive the motion of each Gaussian and render images.

## 3.5 Adaptive Control of Superpoints

We use the farthest point sampling algorithm to sample $M$ Gaussians to initialize the superpoints. Simply making superpoints learnable is not enough to model complex motion patterns. More importantly, we wish to simplify the skeleton after training to ease pose editing by reducing the number of superpoints. Following 3D-GS [2] and SC-GS [48], we develop an adaptive control strategy to prune, densify, and merge superpoints.

**Prune:** To determine whether a superpoint $\boldsymbol{p}_j$ should be pruned, we calculate its overall impact $W_j = \sum_{i \in \tilde{\mathcal{N}}_j} w_{ij}$, where $\tilde{\mathcal{N}}_j = \{i \mid j \in \mathcal{N}_i\}$ is the set of Gaussians whose $K$ nearest neighbors include superpoints $\boldsymbol{p}_j$. When $W_j < \delta_{prune}$, we prune this superpoint as it is of little contribution to the motion of 3D Gaussians.

**Densify:** Two aspects determining whether a superpoint should be split into two superpoints. On one hand, we clone a superpoint when its impact $W_j$ is greater than a threshold $\delta_{clone}$, indicating there is a great amount of Gaussians associated with this superpoint, and cloning such superpoints helps model fine motion. On the other hand, we calculate the weighted Gaussians gradient norm of superpoint $j$ as:

$$g_j = \sum_{i \in \tilde{\mathcal{N}}_j} \frac{w_{ij}}{\sum_{k \in \tilde{\mathcal{N}}_j} w_{kj}} \|\frac{\partial \mathcal{L}}{\partial \boldsymbol{\mu}_i}\|_2^2, \qquad (13)$$

Table 1: Quality comparison of novel view synthesis for the *D-NeRF* dataset.

| Method | Skeleton | PSNR↑ | SSIM↑ | LPIPS↓ | FPS↑ | resolution | Opt. Time |
|---|---|---|---|---|---|---|---|
| D-NeRF [14] | No | 30.48 | 0.9683 | 0.0450 | <1 | $400 \times 400$ | 20.0 hours |
| TiNeuVox-B [5] | No | 32.60 | 0.9783 | 0.0383 | 0.82 | $400 \times 400$ | 28.0 mins |
| Hexplane [49] | No | 29.81 | 0.9683 | 0.0400 | 1.37 | $400 \times 400$ | 11.5 mins |
| K-Plane hybrid [50] | No | 31.02 | 0.9717 | 0.0495 | 0.52 | $400 \times 400$ | 52.0 mins |
| 4D-GS [20] | No | 34.39 | 0.9830 | 0.0190 | 141.37 | $800 \times 800$ | 20.0 mins |
| SP-GS [22] | No | 37.55 | 0.9884 | 0.0137 | 234.83 | $800 \times 800$ | 52.3 mins |
| D-3D-GS [21] | No | 40.11 | 0.9918 | 0.0120 | 42.10 | $800 \times 800$ | 66.0 mins |
| SC-GS [48] | No | 42.98 | 0.9955 | 0.0028 | 123.04 | $400 \times 400$ | 53.3 mins |
| WIM [3] | Yes | 25.21 | 0.9383 | 0.0700 | 0.16 | $400 \times 400$ | 11 hours |
| AP-NeRF [4] | Yes | 30.91 | 0.9700 | 0.0350 | 1.33 | $400 \times 400$ | 150. mins |
| Ours | Yes | 38.80 | 0.9870 | 0.0095 | 103.98 | $800 \times 800$ | 90.6 mins |
| Ours | Yes | 39.23 | 0.9890 | 0.0070 | 110.90 | $400 \times 400$ | 92.5 mins |

Table 2: Quantitative comparison of novel view synthesis on the *Robots* dataset.

| Method | PSNR↑ | SSIM↑ | LPIPS↓ | FPS↑ | resolution |
|---|---|---|---|---|---|
| WIM [3] | 29.11 | 0.9664 | 0.0350 | 0.11 | $512 \times 512$ |
| AP-NeRF [4] | 32.45 | 0.9784 | 0.0202 | 0.89 | $512 \times 512$ |
| Ours | 34.34 | 0.9809 | 0.0187 | 137.76 | $512 \times 512$ |

where $\mathcal{L}$ is the loss function, which demonstrated in Appendix A. We clone the superpoint $\boldsymbol{p}_j$ if $g_j$ is greater than the threshold $\delta_{grad}$.

**Merge:** We merge the superpoints that should belong to the same rigid part. To determine which superpoints should be merged, we first calculate the transformations $\mathbf{T}_j^t \in \mathfrak{se}3$ for all superpoints at all training timestamps. Then, for each pair $(a, b)$ of superpoints, we calculate the average relative transformations:

$$D_{a,b} = \frac{1}{N_t} \sum_t \| \log(\mathbf{T}_b^{t^{-1}} \mathbf{T}_a^t) \|, \tag{14}$$

where $N_t$ is the number of train timestamps, $\log()$ denotes the operation of converting a rigid transformation matrix to a Lie algebra. A small $D_{a,b}$ indicates two superpoints have similar motion patterns. Therefore, we merge two superpoints $\boldsymbol{p}_a$ and $\boldsymbol{p}_b$ when $D_{a,b} < \delta_{merge}$ and $\boldsymbol{p}_b$ is the $K'$-nearest superpoints of $\boldsymbol{p}_a$.

## 4 Experiments

In this section, we present the evaluation of our approach, which achieves excellent view-synthesis quality and real-time rendering speed. We also evaluate the contribution of each component through an ablation study. Additionally, we demonstrate the class-agnostic reposing capability. Please refer to our project `https://dnvtmf.github.io/SK_GS/` for video visualization.

### 4.1 Datasets and Evaluation Metrics

To ensure fair comparison with previous work, we choose the same datasets and configurations as AP-NeRF[4]. Specifically, we choose three multi-view video datasets. First, the *D-NeRF* [14] dataset is a sparse multi-view synthesis dataset, which includes 5 humanoids, 2 other articulated objects, and a multi-component scene. Each scene contains 50-200 frames. We choose 6 of 8 scenes [1] The second dataset, *Robots* [3], contains 7 topologically varied robots with multi-view synthetic video. We use 18 views for training and 2 views for evaluation. The third dataset, *ZJU-MoCap* [34], is commonly used

---

[1]We exclude the *Lego* and *Bouncing Balls* scene due to errors in the *Lego* test split and the multi-component nature of *Bouncing Balls* .

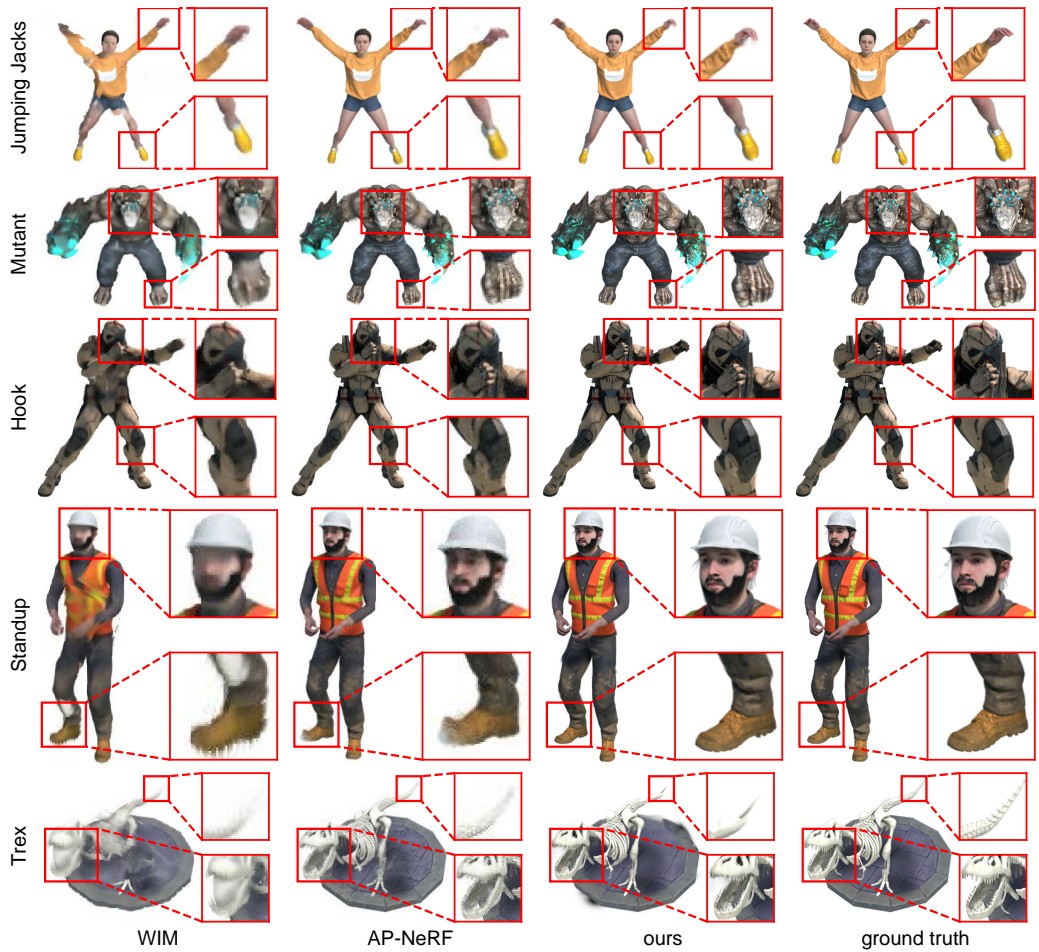

Figure 2: Qualitative comparison on *D-NeRF* datasets.

for dynamic human reconstruction. Following WIM [3] and AP-NeRF [4], we evaluate 5 sequences with 6 training views for each sequence. We use three metrics to evaluate the image quality of the novel view, *i.e.*, peak signal-to-noise ratio (PSNR), structural similarity (SSIM) [51], and learned perceptual image patch similarity (LPIPS) [52].

## 4.2 Implementation Details

We implement our framework using PyTorch. The number of superpoints is initialized as 512. For both deformable field $\Phi$ and $\Psi$, we adopt the architecture of NeRF[1], *i.e.*, 8-layers MLP where each layer employs 256-dimensional hidden fully connected layer and ReLU activation function. We also employ positional encoding for the input coordinates and time. For optimization, we employ the Adam optimizer and use the different learning rate decay schedules for each component: the learning rate about 3D Gaussians is the same as 3D-GS, while the learning rate of other components undergoes exponential decay, ranging from 1e-3 to 1e-5. We conducted all experiments on a single NVIDIA Tesla V100 (32GB). More implementation details are shown in Appendix A.

## 4.3 Baselines

We mainly compare our method to state-of-the-art template-free articulated methods for view synthesis, *i.e.* WIM [3] and AP-NeRF [4]. Besides, we also compare our method with NeRF-based and 3D-GS-based non-articulated methods. D-NeRF [14] extends NeRF to dynamic scenes by warping a static NeRF. TiNeuVox [5] improves the visual quality and training speed by using voxel grids.

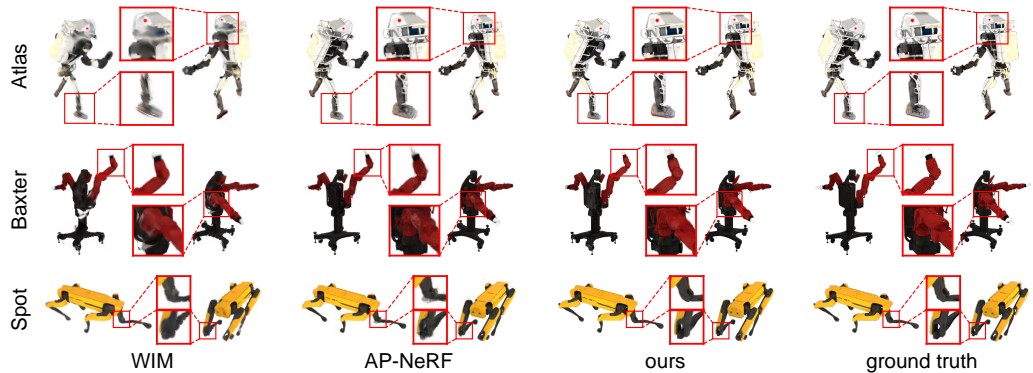

Figure 3: Qualitative comparison for the *Robots*[3] dataset.

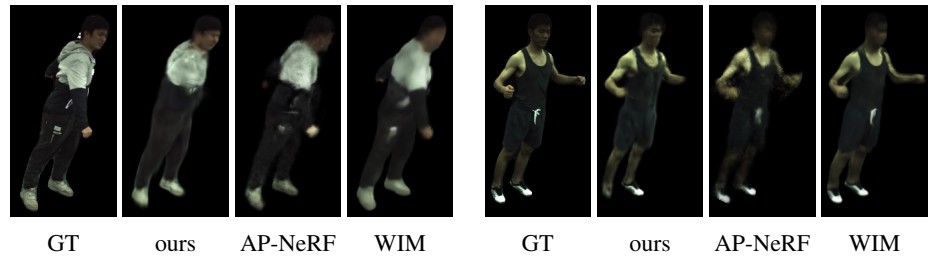

Figure 4: Qualitative comparison for the *ZJU-MoCap* [34] dataset.

HexPlane [49] and K-Planes [50] accelerate NeRF by decomposing the space-time volume into several planes. Similar to D-NeRF, Deformable-3D-GS [21] extends static 3D-GS to the temporal domain. 4D-GS [20] accelerate Deformable-3D-GS by decomposing neural voxel encoding algorithm inspired by HexPlane. Similar to ours, SP-GS [22] and SC-GS [48] employ the superpoints/control points to reconstruct dynamic scenes. However, SP-GS and SC-GS can not extract skeleton from reconstructed model.

### 4.4 Comparisons on Synthetic Dataset

In our experiments, we benchmarked our method against several baselines using the *D-NeRF* dataset and *Robots* datasets. The quantitative comparison results, presented in Tab. 1, demonstrate the superior performance of our approach in terms of both rendering speed and visual quality. Specifically, our method significantly outperforms WIM and AP-NeRF not only in visual quality but also in rendering speed. Compared to 3D-GS based dynamic scenes reconstruction models, our method not only have similar performance, but also discover the skeleton and can repose the object to generate a novel pose. Specifically, the rendering quality of ours is higher than 4D-GS [20] and SP-GS [22], and lower than D-3D-GS [21] and SC-GS [48]. Our method also can achieve real-time rendering (>100 FPS), which is near to the rendering speed of SC-GS [48]. Fig. 2 provides the qualitative comparisons of *D-NeRF* dataset, which demonstrates the advantages of our method over related methods. We also provide results for *Robots* datasets, quantitatively in Tab. 2 and qualitatively in Fig. 3. It is also clear that our approach is capable of producing high-fidelity novel views with real-time rendering speed. Per-scene results are shown in Appendix B.

### 4.5 Comparison on Real-world Dataset

In Tab. 3 and Fig. 4, we compare our method to WIM and AP-NeRF in the *ZJU-MoCap* dataset. We observe that both methods can recover the 3D shape and skeleton models. However, imperfections in the camera calibrations (see Supplement F of [53]), lead to lower visual quality in our results compared to WIM and AP-NeRF. With respect to rendering speed, our approach achieves up to 198.23 FPS. In stark contrast, the rendering speed of WIM and AP-NeRF is extremely slow.

Table 3: Quantitative comparison for the *ZJU-MoCap* [34] dataset.

| Method | PSNR↑ | SSIM↑ | LPIPS↓ | FPS↑ | resolution |
|--------|-------|-------|--------|------|------------|
| WIM[3] | 31.08 | 0.963 | 0.053 | 0.12 | $512 \times 512$ |
| AP-NeRF[4] | 29.60 | 0.958 | 0.063 | 1.31 | $512 \times 512$ |
| ours | 29.11 | 0.961 | 0.063 | 198.23 | $512 \times 512$ |

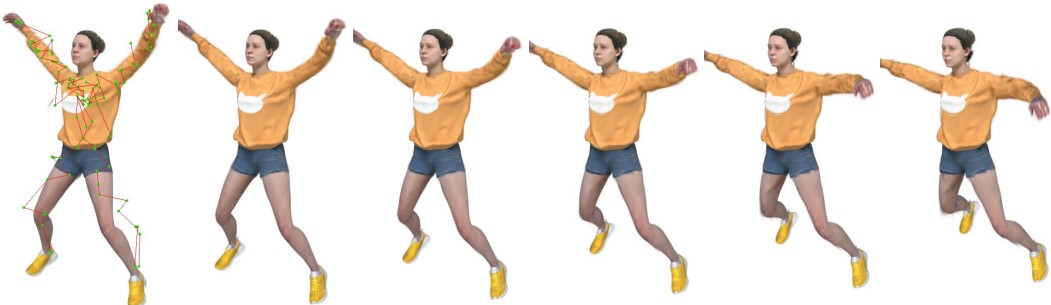

Figure 5: Reposing using skeleton. Interpolation from canonical to novel pose.

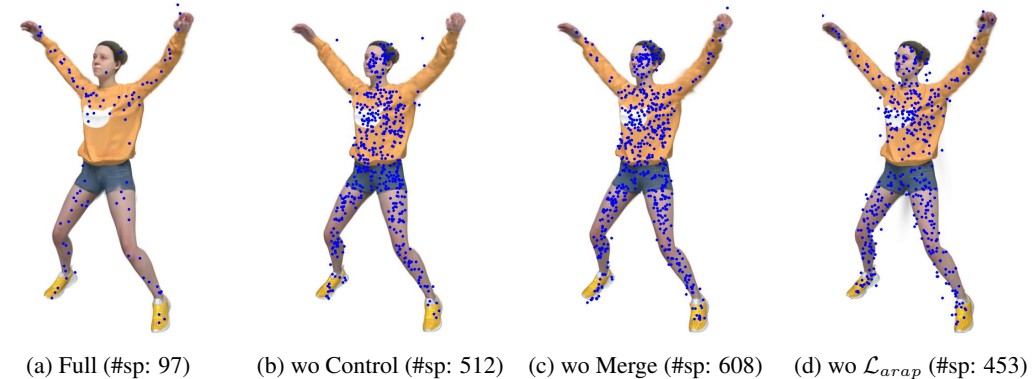

(a) Full (#sp: 97)    (b) wo Control (#sp: 512)    (c) wo Merge (#sp: 608)    (d) wo $\mathcal{L}_{arap}$ (#sp: 453)

Figure 6: We visualize the rendering results of (a) our full method, (b) our method without adaptive control, (c) our method without merge superpoints, (d) our method without $\mathcal{L}_{arap}$ (see Appendix A). #sp denotes the number of superpoints. The blue points denotes superpoints.

## 4.6  Reposing

In Fig. 5, we show our model allows free changes poses and generates animating video by smoothly interpolating the skeletons posing between user-defined poses. Video examples can be found on our project webpage.

## 4.7  Ablation Study

We use superpoints to model the parts and motion of the object. The adaptive control of superpoints is the key to reducing the number of superpoints. Fig. 6 (a), (b) and (c) intuitively illustrate the impact of this control strategy. Without the control strategy, the distribution of superpoints becomes uneven, with sparse representation in the arm region, which negatively impacts motion modeling. The merge process in the control strategy significantly reduces the number of superpoints (97 vs 608), while the superpoints are more distributed in the motion area. Besides, as illustrated in Fig. 6 (d), $\mathcal{L}_{arap}$ (in Appendix A) also plays an important role in controlling the density of superpoints. See Appendix C for more ablation studies.

# 5 Discuss

## 5.1 Limitations

We have demonstrated that our approach can achieve real-time rendering, state-of-the-art visual quality, and straightforward reposing capability by skeleton and kinematic models. However, there are some limitations to our approach. Firstly, similar to WIM and AP-NeRF, the learned skeleton model of our approach is restricted to the kinematic motion space exhibited in the input video. Therefore, the skeleton model may have significant differences from the actual one, and extrapolation to generate arbitrary unseen poses may cause errors. Secondly, our approach has similar limitations as other 3D-GS based methods for dynamic scenes. Specifically, the datasets with inaccurate camera poses will lead to reconstruction failures, and large motion or long-term sequences can also result in failures. Lastly, the paper focuses on building the kinematic model for one articulated object. Exporting build kinematic models for multi-component objects or complex scenes that contain multiple objects remains an opportunity for future research. Additionally, extending this approach to motion capture is an interesting research direction.

## 5.2 Broader Impacts

Although our approach is universal, it is also suitable for rending novel views and poses for humans. Therefore, we acknowledge that our approach can potentially be used to generate fake images or videos. We firmly oppose the use of our research for disseminating false information or damaging reputations.

## 5.3 Conclusion

We have developed a new method for real-time rendering of articulated models for high-quality novel view synthesis. Without any template or annotations, our approach can reconstruct a kinematic model from multi-view videos. With state-of-the-art visual quality and real-time rendering speed, our work represents a significant step towards the development of low-cost animatable 3D objects for use in movies, games, and education.

### Acknowledgements

This work is supported by the Sichuan Science and Technology Program (2023YFSY0008), China Tower-Peking University Joint Laboratory of Intelligent Society and Space Governance, National Natural Science Foundation of China (61632003, 61375022, 61403005), Grant SCITLAB-30001 of Intelligent Terminal Key Laboratory of SiChuan Province, Beijing Advanced Innovation Center for Intelligent Robots and Systems (2018IRS11), and PEKSenseTime Joint Laboratory of Machine Vision.

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

# A  More Implementation Details

The whole optimization progress can be divided into three stages: *dynamic* stage, *joints discovery* stage and *kinematic* stage.

## A.1  Dynamic Stage

In this stage, we aim to reconstruct the appearance of the object, and find the position of underlying joints. Therefore, we employ the $\mathcal{L}_{joint}$, whose formula is:

$$\mathcal{L}_{joint} = \frac{1}{M^2} \sum_{i=1}^{M} \sum_{j=1}^{M} d_{ij} + \frac{1}{M-1} \sum_{(i,j) \in \Gamma} d_{ij} \tag{15}$$

Besides, to obtain a good segmentation of object parts by using superpoints, we employ some regularization losses. To reduce the total number of superpoints, we encourage nearby superpoints to possess same motion patterns via as-rigid-as-possible regularization $\mathcal{L}_{arap}$:

$$\mathcal{L}_{arap} = \sum_{j=1}^{M} \sum_{k \in \mathcal{N}_j^{sp}} \| \log(\mathbf{R}_j^{t\,-1} \mathbf{R}_k^t) \|_2^2 + \| \boldsymbol{o}_j^t - \boldsymbol{o}_k^t \|_2^2. \tag{16}$$

where $\mathcal{N}_j^{sp}$ is the set of $K'$-nearest neighbor superpoints of superpoint $j$. For the blend skinning weight, we employ $\mathcal{L}_{smooth}$ to encourage smoothness by penalizing divergence of skinning weight:

$$\mathcal{L}_{smooth} = \sum_{i=1}^{N} \sum_{j \in \mathcal{N}_i} |w_i - w_j|. \tag{17}$$

Besides, we apply $\mathcal{L}_{sparse}$ to encourage sparsity so that one Gaussian is more likely associated with only one superpoint:

$$\mathcal{L}_{sparse} = - \sum_{i}^{N} \sum_{j}^{B} w_{ij} \log(w_{ij}) + (1 - w_{ij}) \log(1 - w_{ij}) \tag{18}$$

In total, our training loss of *dynamic* stage is:

$$\mathcal{L} = \lambda_0 \mathcal{L}_{rgb} + \lambda_1 \mathcal{L}_{joint} + \lambda_2 \mathcal{L}_{arap} + \lambda_3 \mathcal{L}_{smooth} + \lambda_4 \mathcal{L}_{sparse} \tag{19}$$

where $\lambda = \{1, 1., 10^{-3}, 0.1, 0.1\}$ in our experiments.

In this stage, we conducted training for a total of 40k iterations. Similar to SC-GS [48], our training scheme is as follows:

1. In 0~2k iterations, we fix deformable field $\Phi$.
2. In 2k~10k iterations, we train deformable field $\Phi$ which using the position of Gaussians rather than superpoints as inputs. At 7500 iteration, we sample $M$ Gaussians by using futherest sampling algorithm. At 10k iteration, we initialize superpoints by traind $M$ Gaussians, and re-initialize the Gaussians in canonical space.
3. In $10 \sim 13$k iterations, we fix deformable field $\Phi$.
4. In $13$k $\sim 40$k iterations, all parameters are optimized.

During the first 20k iterations, We do not discover joints and skeleton, *i.e.*, $\lambda_1 = 0$. During 20k $\sim$ 40k iterations, we update the structure of skeleton every 100 iterations. Note that we stopped the gradient between joints and other parts so that the learning of joints has no effect on the learning of Gaussians, superpoints and deformation field.

During training, we adopt the same adaptive control strategy for Gaussians as 3D-GS[2] In details, in iterations 1~7500 and 10k~ 35k, we densify and prune Gaussians every 100 iterations, while the densify grad threshold is set to 0.0002 and the opacity threshold for prune is set to 0.005. We reset the opacity of Gaussians every 3k steps.

We densify and prune superpoints every 1000 steps between iterations 20k and 30k, and the hyper-parameter $\delta_{grad} = 0.0002$ and $\delta_{prune} = 0.001$. We merge superpoints every 1000 steps between iterations 30k and 40k, while threshold $\delta_{merge} = 0.0005$.

## A.2 Skeleton Discovery Stage

In this stage, we aim to initialize deformable field $\Psi$ the using the learned deformable field $\Phi$ in *dynamic* stage. Firstly, we fix skeleton structure $\Gamma$, and initialize the joints $\boldsymbol{j}_k, k = 1, \ldots, M - 1$ by underlying joints $\boldsymbol{j}_{ab}$ according to $\Gamma$. We then cache the outputs of deformable field $\Phi$ for all timestamps in the training dataset. Next, we use cached motions of superpoints to optimize the parameters of deformable field $\Psi$, the positon of the joints $\boldsymbol{j}_k$ by employing Adam optimizer and following loss functions:

$$\mathcal{L}_{discovery} = \frac{1}{M} \sum_{i=1}^{M} \lambda_5 (\|\boldsymbol{\mu}_i^t - \hat{\boldsymbol{\mu}}_i^t\|_2^2 + \| \log(\mathbf{R}_i^{t-1} \hat{\mathbf{R}}_i^t)) + \lambda_6 \|(\mathbf{R}_i^t \boldsymbol{p} + \boldsymbol{\mu}^t) - (\hat{\mathbf{R}}_i^t \boldsymbol{p} + \hat{\boldsymbol{\mu}^t})\| \quad (20)$$

where $\hat{\boldsymbol{\mu}}_i^t$ and $\hat{\mathbf{R}}_i^t$ is translations and rotations calculated by deformable field $\Phi$, while $\boldsymbol{\mu}_i^t$ and $\mathbf{R}_i^t$ is translations and rotations calculated by deformable field $\Psi$ and kinematic model. $\lambda_5 = 1$ controls the weights of the transform matrix difference, and $\lambda_6 = 0.1$ adjusts the weights of relative offset of the superpoints position at timestamp $t$.

For this stage, we train for 10k iterations with a fixed learning rate of $10^{-3}$.

## A.3 Kinematic Stage

In the *kinematic* stage, we optimize the kinematic model, joints and Gaussians, while keeping the number of Gaussians, the number of superpoints and the structure of skeleton fixed. We also conduct training for a total of 40k iterations by only using $\mathcal{L}_{rgb}$.

## A.4 Others

For the deformable field $\Phi$ and $\Psi$, we adopt the following positional encoding for the input coordinates $\boldsymbol{x} \in \mathbb{R}^{N \times 3}$ and time $t \in [0, 1]$:

$$\gamma(p) = (\sin(2^k p), \cos(2^k p))_{k=0}^L, \quad (21)$$

where $L = 10$ for $\boldsymbol{x}$ and $L = 6$ for $t$.

The initial number of superpoints is set to 512, while $K = K' = 5$.

# B  Per Scene Results

## B.1  Per-Scene Results for *D-NeRF* dataset

Tab. 4 shows the per-scene quantitative results for the *D-NeRF* dataset.

## B.2  Per-scene Results for *Robots* dataset

Tab. 5 shows the per-scene quantitative results for *Robots* dataset.

# C  More Results

## C.1  Required Resources

As shown in Tab. 6, we present detailed information on the optimization time and the required resources.

## C.2  Ablation study for the number of initial superpoint

The ablation study on the number of initial superpoint $M$ is shown in Tab. 7.

As shown in the Fig. 7, our method struggles to accurately reconstruct the endpoints of objects.

| PSNR↑ | JumpingJacks | Mutant | Hook | T-Rex | StandUp | HellWarrior | Average |
|---|---|---|---|---|---|---|---|
| D-NeRF [14] | 32.80 | 31.29 | 29.25 | 31.75 | 32.79 | 25.02 | 30.48 |
| TiNeuVox-B [5] | 34.23 | 33.61 | 31.45 | 32.7 | 35.43 | 28.17 | 32.60 |
| Hexplane [49] | 31.65 | 33.79 | 28.71 | 30.67 | 34..6 | 24.24 | 29.81 |
| K-Plane hybrid [50] | 32.64 | 33.79 | 28.5 | 31.79 | 33.72 | 25.7 | 31.02 |
| D-3D-GS [21] | 37.59 | 42.61 | 37.09 | 37.67 | 44.30 | 41.41 | 40.11 |
| 4D-GS [20] | 35.44 | 37.43 | 33.01 | 33.61 | 38.11 | 28.77 | 34.39 |
| SP-GS [22] | 35.56 | 39.43 | 35.36 | 32.69 | 42.07 | 40.19 | 37.55 |
| SC-GS [48] | 41.62 | 45.08 | 39.81 | 40.70 | 47.81 | 42.88 | 42.98 |
| WIM [3] | 29.77 | 25.80 | 25.33 | 26.19 | 27.46 | 16.71 | 25.21 |
| AP-NeRF [4] | 34.50 | 28.56 | 30.24 | 32.85 | 31.93 | 27.53 | 30.94 |
| Ours ($800 \times 800$) | 36.95 | 40.95 | 36.64 | 35.10 | 43.52 | 39.65 | 38.80 |
| Ours ($400 \times 400$) | 36.70 | 41.96 | 36.77 | 36.14 | 43.84 | 39.95 | 39.23 |

| SSIM↑ | JumpingJacks | Mutant | Hook | T-Rex | StandUp | HellWarrior | Average |
|---|---|---|---|---|---|---|---|
| D-NeRF [14] | 0.98 | 0.97 | 0.96 | 0.97 | 0.98 | 0.95 | 0.9683 |
| TiNeuVox-B [5] | 0.98 | 0.98 | 0.97 | 0.98 | 0.99 | 0.97 | 0.9783 |
| Hexplane [49] | 0.97 | 0.98 | 0.96 | 0.98 | 0.98 | 0.94 | 0.9683 |
| K-Plane hybrid [50] | 0.977 | 0.983 | 0.954 | 0.981 | 0.983 | 0.952 | 0.9717 |
| D-3D-GS [21] | 0.9929 | 0.987 | 0.9858 | 0.995 | 0.9947 | 0.9953 | 0.9918 |
| 4D-GS [20] | 0.9857 | 0.988 | 0.9760 | 0.985 | 0.9898 | 0.9733 | 0.9830 |
| SP-GS [22] | 0.9950 | 0.9868 | 0.9804 | 0.9861 | 0.9926 | 0.9894 | 0.9884 |
| SC-GS [48] | 0.9957 | 0.9977 | 0.9934 | 0.9972 | 0.9981 | 0.9908 | 0.9955 |
| WIM[3] | 0.97 | 0.95 | 0.94 | 0.94 | 0.96 | 0.87 | 0.9383 |
| AP-NeRF[4] | 0.98 | 0.96 | 0.97 | 0.98 | 0.97 | 0.96 | 0.9700 |
| Ours ($800 \times 800$) | 0.9883 | 0.9921 | 0.9847 | 0.9877 | 0.9933 | 0.9761 | 0.9870 |
| Ours ($400 \times 400$) | 0.9889 | 0.9951 | 0.9869 | 0.9934 | 0.9950 | 0.9749 | 0.9890 |

| LPIPS ↓ | JumpingJacks | Mutant | Hook | T-Rex | StandUp | HellWarrior | Average |
|---|---|---|---|---|---|---|---|
| D-NeRF[14] | 0.03 | 0.02 | 0.11 | 0.03 | 0.02 | 0.06 | 0.0450 |
| TiNeuVox-B[5] | 0.03 | 0.03 | 0.05 | 0.03 | 0.02 | 0.07 | 0.0383 |
| Hexplane[49] | 0.04 | 0.03 | 0.05 | 0.03 | 0.02 | 0.07 | 0.0400 |
| K-Plane hybrid[50] | 0.0468 | 0.0362 | 0.0662 | 0.0343 | 0.031 | 0.0824 | 0.0495 |
| D-3D-GS[21] | 0.0126 | 0.0052 | 0.0144 | 0.0098 | 0.0063 | 0.0234 | 0.0120 |
| 4D-GS[20] | 0.0128 | 0.0167 | 0.0272 | 0.0131 | 0.0074 | 0.0369 | 0.0190 |
| SP-GS [22] | 0.0069 | 0.0164 | 0.0187 | 0.0243 | 0.0096 | 0.0066 | 0.0137 |
| SC-GS [48] | 0.0030 | 0.0011 | 0.0037 | 0.0014 | 0.0008 | 0.0068 | 0.0028 |
| WIM[3] | 0.04 | 0.06 | 0.06 | 0.08 | 0.04 | 0.14 | 0.0700 |
| AP-NeRF[4] | 0.03 | 0.03 | 0.05 | 0.02 | 0.02 | 0.06 | 0.0350 |
| Ours ($800 \times 800$) | 0.0086 | 0.0038 | 0.0089 | 0.0105 | 0.0045 | 0.0203 | 0.0095 |
| Ours ($400 \times 400$) | 0.0090 | 0.0022 | 0.0073 | 0.0054 | 0.0026 | 0.0155 | 0.0070 |

| FPS↑ | JumpingJacks | Mutant | Hook | T-Rex | StandUp | Hellwarrior | Average |
|---|---|---|---|---|---|---|---|
| D-3D-GS [21] | 16.35 | 102.51 | 32.3 | 20.59 | 49.14 | 31.68 | 42.10 |
| 4D-GS [20] | 112.89 | 129.59 | 147.38 | 144.46 | 152.46 | 161.41 | 141.37 |
| SP-GS [22] | 271.27 | 210.42 | 230.35 | 186.65 | 260.34 | 249.93 | 234.83 |
| SC-GS [48] | 128.05 | 122.01 | 129.81 | 105.24 | 132.91 | 120.24 | 123.04 |
| WIM [3] | 0.19 | 0.17 | 0.13 | 0.16 | 0.18 | 0.13 | 0.16 |
| AP-NeRF [4] | 1.57 | 1.42 | 1.11 | 1.34 | 1.48 | 1.06 | 1.33 |
| Ours ($800 \times 800$) | 106.81 | 101.04 | 103.60 | 98.38 | 102.02 | 112.03 | 103.98 |
| Ours ($400 \times 400$) | 109.61 | 104.22 | 109.11 | 109.15 | 110.03 | 123.28 | 110.90 |

Table 4: Quantitative Results Per-Scene in *D-NeRF* dataset.

| PSNR↑ | Atlas | Baxter | Cassie | Iiwa | Nao | Pandas | Spot | Average |
|---|---|---|---|---|---|---|---|---|
| WIM[3] | 25.01 | 23.36 | 30.08 | 32.77 | 28.68 | 35.93 | 27.92 | 29.11 |
| AP-NeRF[4] | 32.56 | 30.74 | 31.44 | 35.78 | 30.64 | 32.44 | 33.57 | 32.45 |
| ours | 34.89 | 31.96 | 32.08 | 39.34 | 33.24 | 32.02 | 36.85 | 34.34 |

| SSIM↑ | Atlas | Baxter | Cassie | Iiwa | Nao | Pandas | Spot | Average |
|---|---|---|---|---|---|---|---|---|
| WIM[3] | 0.9405 | 0.9538 | 0.9712 | 0.9866 | 0.9546 | 0.9878 | 0.9704 | 0.9664 |
| AP-NeRF[4] | 0.9833 | 0.9759 | 0.9775 | 0.9884 | 0.9615 | 0.9785 | 0.9837 | 0.9784 |
| ours | 0.9891 | 0.9561 | 0.9792 | 0.9928 | 0.9741 | 0.9896 | 0.9856 | 0.9809 |

| LPIPS↓ | Atlas | Baxter | Cassie | Iiwa | Nao | Pandas | Spot | Average |
|---|---|---|---|---|---|---|---|---|
| WIM[3] | 0.0648 | 0.0455 | 0.0359 | 0.0160 | 0.0368 | 0.0177 | 0.0283 | 0.0350 |
| AP-NeRF[4] | 0.0170 | 0.0196 | 0.0281 | 0.0129 | 0.0323 | 0.0203 | 0.0110 | 0.0202 |
| ours | 0.0092 | 0.0430 | 0.0249 | 0.0062 | 0.0187 | 0.0172 | 0.0114 | 0.0187 |

| FPS↑ | Atlas | Baxter | Cassie | Iiwa | Nao | Pandas | Spot | Average |
|---|---|---|---|---|---|---|---|---|
| WIM[3] | 0.08 | 0.08 | 0.10 | 0.17 | 0.07 | 0.10 | 0.08 | 0.10 |
| AP-NeRF[4] | 0.66 | 1.26 | 0.33 | 0.35 | 1.02 | 0.39 | 1.73 | 0.82 |
| ours | 123.32 | 149.43 | 139.08 | 122.59 | 141.58 | 146.73 | 141.62 | 137.76 |

Table 5: Quantitative Results Per-Scene in *Robots* dataset.

Table 6: Optimization time and required resources for each scene in the *D-NeRF* dataset.

| scene | hellwarrior | hook | jumpingjacks | mutant | standup | trex | average |
|---|---|---|---|---|---|---|---|
| Training Time (h) | 1.17 | 1.52 | 1.50 | 1.55 | 1.40 | 1.92 | 1.51 |
| GPU VRAM(GB) | 1.23 | 3.63 | 2.31 | 3.33 | 2.19 | 4.32 | 2.83 |
| num. of Gaussians ($\times 10^5$) | 0.60 | 1.54 | 1.11 | 1.44 | 0.97 | 1.92 | 1.28 |
| num. of superpoints | 188 | 184 | 112 | 51 | 134 | 42 | 118.5 |

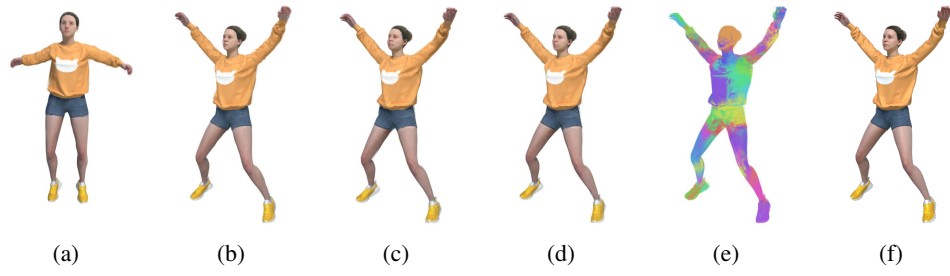

|      |      |      |      |      |      |
|------|------|------|------|------|------|
| (a)  | (b)  | (c)  | (d)  | (e)  | (f)  |

Figure 8: Compare rendered images between canonical space and the warp space of timestamp 0. (a) canonical space in *Dynamic* stage, (b) at time 0 in *Dynamic* stage, (c) canonical space in *Kinematic* stage, (d) at time 0 in *Kinematic* stage, (e)LBS at time 0 in *Kinematic* stage (f)ground truth at time 0.

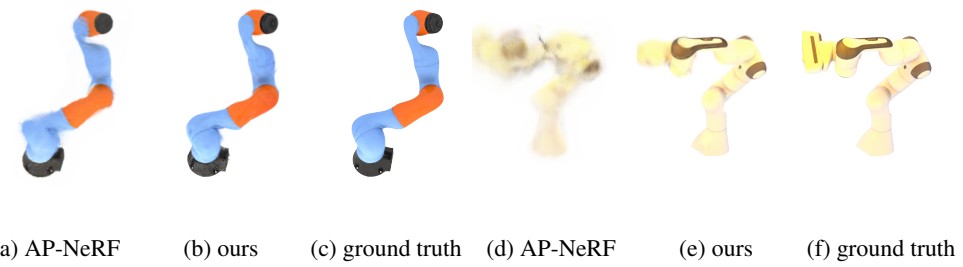

| (a) AP-NeRF | (b) ours | (c) ground truth | (d) AP-NeRF | (e) ours | (f) ground truth |
|---|---|---|---|---|---|

Figure 7: Compare to AP-NeRF, ours method are more robust for complex motion.

Table 7: Ablation study for the number of initial superpoint $M$ on the 'hellwarrior' scene of *D-NeRF* dataset.

| $M$ | 128 | 256 | 384 | 512 | 640 | 768 | 896 | 1024 |
|---|---|---|---|---|---|---|---|---|
| PSNR↑ | 39.61 | 39.70 | 39.83 | 39.69 | 39.89 | 39.56 | 39.72 | 40.43 |
| SSIM↑ | 0.9765 | 0.9766 | 0.9779 | 0.9773 | 0.9781 | 0.9766 | 0.9767 | 0.9800 |
| LPIPS↓ | 0.0191 | 0.0179 | 0.0173 | 0.0183 | 0.0169 | 0.0183 | 0.0186 | 0.0155 |

