# OpenReview forum: "Template-free Articulated Gaussian Splatting for Real-time Reposable Dynamic View Synthesis"
_NeurIPS.cc/2024/Conference — NeurIPS 2024 poster_

### Official Review · Reviewer_XcTH · 2024-06-28

**Soundness:** 3
**Presentation:** 4
**Contribution:** 3
**Rating:** 7
**Confidence:** 4

**Summary:**

The authors propose to combine a reposable 4D reconstruction from multi-view video based on a skeletal LBS model with 3D Gaussian splatting. To this goal they introduce a novel strategy for estimation of the skeletal model from a superpoint clustering. The results demonstrate a superior image quality and, thanks to the representation, also fast rendering.

**Strengths:**

S1) Implementation details provided for reproducibility.

S2) The claims are validated on common datasets.

S3) The result quality is visibly better than the prior work.

S4) The skeleton construction is novel, technically sound and produces reposable models in variety of scenes.

S5) The quality of exposition is good.

**Weaknesses:**

W1) The skeletons are over-segmented and unnatural and likely would not be very friendly for a human animator. They may still be suitable for data fitting but do not provide nearly as much regularization as an "optimal" (ground truth) skeleton would.

W2) The limitations and broader impacts are only discussed in the Appendix which I do not see as a responsible practice. It suggests that the authors do not give downsides of the method the same importance as to the upsides.

W3) The authors claim to report Statistical Significance without further comments (checklist item #7), but I cannot see any such features in the paper.

W4) It may be a good idea to consider a higher quality captured dataset than ZJU-Mocap. It does not seem to allow for a useful comparison between the methods.


Other minor issues and suggestions:

- Figure 1: Each superpoints -> superpoint

- L145: related rotation matrix -> relative?

- $\mathbf{W}$ is overloaded in Eq. 1 and Eq. 6 for two distinct things which is not ideal.

- Eq. 13: $\mathcal{L}$ without suffix is not defined.


------------------------
**Justification of recommendation**
A solid paper with its incremental but non-trivial contribution stemming mainly from the novel skeleton construction. The experimental results are convincing and the main downside is the clutter and complexity of the recovered skeleton. Despite this, I am currently comfortable recommending acceptance under the assumption that the exposition issues are addressed (especially limitations). My final decision might change based on the rebuttal.

**Questions:**

Q1) The Eq. 13 could use more discussion. Does the formulation avoid over-segmentation of large but correctly rigid parts to multiple sub-parts? It would be useful to see the LBS visualized for the final shapes.

Q2) Why are the skeletons so complex and noisy? Is that perhaps related to my other question about over-segmentation in Eq. 13? Are such skeletons practical for re-animation? How was re-animation done in the video? How many parameters / joint motions had to be defined?

Q3) Why are there different resolutions reported for each method (Table 2 & 3). Does it mean each method was validated against a different reference image resolution? Or does it mean the methods were all tested against a full resolution reference but some were only trained using low-resolution data? That could potentially introduce considerable bias.

Q4) The authors also do not provide any failure case examples/images. Does the method work perfectly for all tested scenes? What about the robots from the limitation Figure 6 of AP-NeRF? Does the new method handle these cases better?

**Limitations:**

The limitations and broader impacts are only discussed in the Appendix which I do not see as a responsible practice. It suggests that the authors do not give downsides of the method the same importance as to the upsides.

---

> ### Author Rebuttal · Authors · 2024-08-07
>
> Thank you for the detailed comments.
>
> - **Limitations/W2**: Limitations and broader impacts
>
>   **Answer**: We will move the limitations and broader impacts to the main paper in the final version.
>
> - **Q1**: More discussion about  Eq. 13.
>
>   **Answer**: Based $g_i$ calculated by Eq. 13, we can densify more superpoints to accommodate complex motions. Therefore, in theory, this formula will not split a large rigid part into multiple sub-parts. Even if more sub-parts are generated, since the motion of these sub-parts is the same, the *merging process* will merge the sub-parts into one again.
>
> - **Q2**: Why are the skeletons so complex and noisy?
>
>   **Answer**: To re-animate (/re-pose) the $k$-th joint of the object, we simply apply an additional rotation matrix $\Delta \mathbf{R}_k \in \mathrm{SO}(3)$ on the rotation $\hat{\mathbf{R}}_k$ predicted by deformable filed $\Psi$ (Eq. 11), i.e., $\hat{\mathbf{R}}_k' = \Delta \mathbf{R}_k \hat{\mathbf{R}}_k$. Compared with 6 DoF of superpoints in the Dynamic stage, each joint in the Kinematic Stage has only 3 DoF i.e., only can rotate but not translate. Therefore, the extra DoF of superpoints is the main reason are the main reason why the discovered skeletons are so complex and noisy. As shown in Fig. 1 in the attached `PDF`, there are some inconsistencies in the 3D model in canonical space of Dynamic stage. However, the rendered image in the given timestamp matches the ground truth. The price of achieving all this is that the skeleton is complex and noisy.
>
>   Another reason is that reconstucted motion of superpoint is imperfect. The number of superpoints is therefore greater than the number of physically rigid parts. As a result, the skeleton discovered by the motion of superpoints  becomes complex and noisy.
>
> - **Q3**: Why are there different resolutions reported for each method?
>
>   **Answer**: To solve this concern, we present the results of our method with the same resolutions in Table 1 in the attached `PDF`. The training and testing resolutions are the same. For example, when AP-NeRF is trained on $400\times 400$ images, it is also tested on $400 \times 400$ images.
>
> - **A4**: Any failure case examples/images
>
>   **Answer**: For scenes with complex motions (e.g., combining long chains of rotations with texture cue ambiguity), our method can achieve better performance than AP-NeRF. Figure 2 in the attached `PDF` compares our method with AP-NeRF for the failure case of AP-NeRF. As shown in the figure, our method may not be able to reconstruct the endpoints of objects well.
>
> - **W3**: Statistical Significance
>
>   **Answer**: The answer of Statistical Significance is No. And the Justification is: We do not report error bars or other information about statistical significance as this is not a common procedure in the field and does not contribute to understanding our evaluation.
>
> - **W4**: Higher quality captured dataset.
>
>   **Answer**: Thanks for your suggestion. We follow the experiment setting of AP-NeRF. We will evaluate our method on higher-quality captured datasets in the future.

---

> > ### Comment · Reviewer_XcTH · 2024-08-08
> > **Thank you**
> >
> > Thank you for the rebuttal, I have no further questions at the moment.
> >
> > The visualization of the LBS weights in the PDF is useful and while it rather shows limited physiological plausibility of the animation tree, it would be good to include it in the paper to visualize any such limitations.

---

### Official Review · Reviewer_m4oH · 2024-07-09

**Soundness:** 3
**Presentation:** 3
**Contribution:** 3
**Rating:** 6
**Confidence:** 4

**Summary:**

This paper presents a novel approach for learning articulated objects, alongside the skeletons/kinematic trees directly from input videos, eliminating the need for pre-defined meshes or hand-crafted skeleton priors.

Specifically, the paper introduces a hierarchical 3D Gaussian representation, where a set of superpoints are used as guidance for deforming the sub-points using linear blend skinning (LBS). The skeletal structure is further derived from the superpoints, providing articulated structures that enable explicit pose control. Rendering is done by Gaussian Splatting, enabling real-time performance.

Overall, the paper tackles a very challenging and useful problem for 3D modeling, the manuscript is easy to follow, and the approach is interesting.

**Strengths:**

The strengths of this paper lie in how it brilliantly leverages 3D Gaussians for capturing the underlying articulated structures in a video, without the need for 3D annotations or pre-defined structure priors.

 The design of superpoints naturally models prominent candidates that can serve as control points. Furthermore, as the control points are “learned” automatically, it can potentially arrive at a representation that better suits the possible motion of the articulated subject.

Overall, the approach presented in the paper is pretty neat, and the experiments show pretty promising qualitative and quantitative results.

**Weaknesses:**

The approach does have some room for improvement. Specifically,
- Limited reposability. As mentioned in L372-373, the approach is limited to the motion space in the input video. It would be great if the papers could include visual results for these failure cases. It will be interesting to see how good the learned LBS weights are.
- Evaluated on datasets with limited motions: the videos used in the paper mostly contain repetitive motion sequences, and/or with small motions. It will be interesting to see how the proposed method performs on videos with complex/diverse/large motions (e.g., AIST datasets). Also, it is similarly unclear how the method can perform on in-the-wild videos with uncontrolled lighting, or with only a single view.

Overall, these weaknesses are very common among template-free approaches, not specifically to the proposed method itself. Nevertheless, it would be great if the paper could include more figures, visual results, and analysis regarding these cases.

There are also some issues regarding the experiments, which I detailed in the Questions section below.

**Questions:**

Some comments regarding the evaluation sections:
- Tab 1, 2: does the performance gain come mainly from using a higher resolution, or does it come from “capturing better articulated structures”? While Gaussian splatting enables us to use higher resolution due to its rendering speed, it would be great if the paper could also include results with resolutions comparable to other setting (400x400 in Tab 1, and 800x800 in Tab 2).
- Is there a way to properly evaluate how good the learned skeleton structure is? E.g., training a skeleton-based 3D articulated model using the skeleton from AP-NeRF v.s. WIM v.s. the proposed method.

Also, one small issue:
- L147: should be (R^t_b)^-1 instead of R^t-1_b

**Limitations:**

The paper discussed some of their limitations, but it would be great if the paper could include more analysis/visual results for the issues mentioned above in the Weaknesses section.

---

> ### Author Rebuttal · Authors · 2024-08-06
>
> We thank the reviewer for the useful suggestions.
>
> - **Q1**: Higher resolution
>
>   **Answer**: We provide the results at the same resolution as WIM and AP-NeRF in the attached `PDF`. We believe the improved performance mainly comes from the powerful representation ability of 3D-GS and the better capture of articulated structures.
>
> - **Q2**: Evaluate skeleton model
>
>   **Answer**: The biggest challenge in evaluating skeleton models is that the learned skeleton is uncertain in terms of the number of joints and the connections between joints.
>
>   In our view, there are three imperfect methods that might be used to evaluate the quality of the learned skeleton structure.
>   1. As shown in WIM, we can first learn a mapping between the learned skeleton model and a template (e.g., SMPL for humans), and then evaluate the mapped template. However, this way requires a template and the mapping process will introduce more errors.
>   2. Given test images with camera poses, all parameters except the rotation of joints are fixed, and the learned model is then optimized to fit the test images. After test-time optimization, the difference between the rendered image and test images shows how good the learned skeleton structure is. However, test-time optimization may fail or be suboptimal.
>   3. The repose process can be treated as an image generating process (e.g., GAN, Diffusion model). Therefore, we evaluate the learned skeleton structure by using the metrics (e.g., FID) that are common in the generative model. However, obtaining the rotation range of each joint and test camera pose is challenging.
>
>   Overall, rigorously evaluating skeleton models is a challenging problem, and we will try to address in future work.

---

> > ### Comment · Reviewer_m4oH · 2024-08-11
> > **Thanks for the detailed responses**
> >
> > After going through the author's responses, as well as the comments from fellow reviewers. The experimental results, along with the rebuttal responses, adequately support the claim in the paper, and I appreciate how the work neatly combines existing techniques/concepts to capture the 3D appearance and, most importantly, the motion structure. Therefore, I stand by my original rating -- Weak Accept.

---

### Official Review · Reviewer_57cM · 2024-07-10

**Soundness:** 2
**Presentation:** 3
**Contribution:** 2
**Rating:** 5
**Confidence:** 4

**Summary:**

The paper introduces a method combining 3D Gaussian Splatting and superpoints for dynamic object modeling, achieving real-time rendering and high visual fidelity. Empirical results show that the proposed method achieves state-of-the-art results on several benchmarks.

**Strengths:**

1. The paper is well-written and easy to follow. The main contribution and methodology are well illustrated.
2. The use of an adaptive control strategy to manage superpoints is innovative and helps in optimizing the model, avoiding redundancy, and maintaining efficiency.

**Weaknesses:**

1. Although this paper achieves real-time rendering compared to AP-NERF, I find it somewhat incremental and lacking in innovation since most parts of the method are existing concepts.
2. This paper emphasizes the concept of "Reposable," but the related experiments are very limited. A thorough analysis of this aspect could effectively distinguish this paper from AP-NERF.
3. This method compares fewer baselines, and the quantitative results do not show significant improvements in rendering effects and speed compared to the baselines, as shown in Table 3, Table 4, and Table 5.

**Questions:**

1. The number of $M$ is not given in the paper, the author should do an ablation study on it.
2. The proposed method and results should be discussed with the latest relevant methods as referenced in [1].

[1] SC-GS: Sparse-Controlled Gaussian Splatting for Editable Dynamic Scenes.

**Limitations:**

Please refer to the Weaknesses and Questions.

---

> ### Author Rebuttal · Authors · 2024-08-06
>
> We sincerely thank you for your time and efforts.
>
> - **W1**: Lacking in innovation
>
>   **Answer**: While our work builds upon previous works, to the best of our knowledge, it is the first work to discover the skeleton of articulated objects represented by 3D Gaussian Splatting.
>
> - **W2**: Distinguish this paper from AP-NERF by a thorough analysis
>
>   **Answer**: Both our method and AP-NeRF are proposed for learning articulated representations of dynamic objects. However, there are two main differences between our method and AP-NeRF:
>    1. While AP-NeRF is based on point-based NeRF representations, our method is based on 3D Gaussian Splatting, which is key to achieving real-time rendering.
>    2. While AP-NeRF extracts the skeleton using the Medial Axis Transform (MAT), our method discovers the skeleton based on the motion of superpoints.
>
> - **W3**: This method compares fewer baselines, and the quantitative results do not show significant improvements
>
>   **Answer**:
>   - Unlike other methods focused on reconstructing dynamic scenes (e.g., D-NeRF, HyperNeRF, Deformable-3D-GS, 4D-GS), our work, alongside with WIM and AP-NeRF, uniquely addresses explicit skeleton extraction from videos without any priors.
>   -  As evidenced in Tables 3, 4, and 5, our approach achieves real-time rendering (>100 FPS), surpassing WIM and AP-NeRF (<1 FPS). Furthermore, our method exhibits superior rendering quality on the D-NeRF and Robots datasets.
> - **Q1**:  Ablation study for the number of $M$
>
>   **Answer**: $M$ is the number of superpoints, which is initialized as 512 as shown in L212. We conduct the ablation study for $M$. The results are shown in Table 2 in the attached `PDF`.
>
> - **Q2**: Comparison with SC-GS
>
>   **Answer**: Quantitative comparison between our method and SC-GS is provided in Table 1 in the attached `PDF`. Besides,
>   - **Difference**: Our method extracts the explicit skeleton model and refines it during the *Kinematic* stage, while SC-GS does not.
>   - **Similarity**: Both our method and SC-GS utilize the sparse points (i.e., superpoints in ours, control points in SC-GS) with LBS.  Notably, the Dynamic stage in our approach can be replaced with SC-GS.
>   - **Difference**: While SC-GS employs the Gaussian-kernel RBF to compute the weights of LBS, our method directly learns it (i.e., $\mathbf{W}$ in Eq 7).
>   - **Difference**: SC-GS can edit motion by minimizing APAR energy. Compared with SC-GS, our method directly manipulates the skeleton, which is more efficient and simpler.

---

> > ### Comment · Reviewer_57cM · 2024-08-12
> >
> > Thank you for the detailed response. I have carefully read your reply but still have some concerns.
> >
> > Firstly, regarding the novelty of the work, while it may be the first to propose the concept of a skeleton in 3DGS, SC-GS has already introduced the use of control points in 3DGS, although it did not explicitly calculate the connections. This significantly impacts the novelty of the work.
> >
> > Secondly, the difference in LBS design mentioned by the authors as a distinction from SC-GS seems to be a relatively minor modification.
> >
> > Lastly, and more critically, in the PDF provided by the authors, there is a significant performance gap between the proposed method and SC-GS, which makes me question the rationale behind these design differences.
> >
> > In conclusion, I remain negative about the work's overall contribution.

---

> ### Author Response · Authors · 2024-08-12
> **Thank you for your feedback**
>
> First of all, although SC-GS uses control points to implement motion editing, there are no constraints between the control points. Therefore, compared to re-pose with skeleton, the motion editing in SC-GS :
> -  1) is easy to generate physically unrealistic motion, and thus render unrealistic images;
> -  2) need an additional optimization process, which is more complex and time-consuming.
>
> We believe that discovering the skeleton of an object will be helpful in certain fields, such as game production, video generation, physical simulation, etc.
>
> Second, it must be pointed out that the design of LBS is **not** the contribution of our paper. Our main contribution is to discover the skeleton of an articulated object by utilizing the motion of superpoints.
>
> Third, the main reasons for the performance gap are as follows:
> - To simplify the skeleton, the proposed adaptive control strategy significantly reduces the number of superpoints. ($512\to \sim 100$). The number of superpoints has an impact on performance.
> - According to the code of SC-GS, SC-GS uses more tricks than ours, such as longer training time (80k), an extra MLP for input time (called time-net), additional hyper-features to help k-nearest neighbor search, the initialization train for control points and so on.
>
> We are incorporating SC-GS into our work to achieve even higher performance.
>
> If you have any further questions or would like additional clarification, please do not hesitate to contact us. We would be more than happy to provide additional information or discuss any aspect of our work in greater detail. Your feedback is deeply appreciated, and we remain fully committed to addressing any concerns you may have.

---

> > ### Comment · Reviewer_57cM · 2024-08-12
> >
> > Thanks for your response.  I understand that the rebuttal time is limited, and I also acknowledge the technical differences between this work and SC-GS.
> >
> > However, I believe the overall motivation remains the same, which is to incorporate articulated information into 3DGS to control dynamic generation. The potential issues with SC-GS mentioned by the authors, as well as the performance gap, are not supported by any experimental evidence, so they fail to convince me.
> >
> > Additionally, if the focus of the paper is on obtaining the object's skeleton, there are many methods, such as [1], [2], and [3], that can extract heterogeneous skeletons from videos or meshes.
> >
> > Finally, I will maintain my score unless there is additional evidence that can prove the points raised by the authors.
> >
> > [1] CASA: Category-agnostic Skeletal Animal Reconstruction, NIPS 2022.
> >
> > [2] Object Wake-Up: 3D Object Rigging from a Single Image, ECCV 2022.
> >
> > [3] RigNet: Neural Rigging for Articulated Characters, TOG 2020.

---

> > > ### Author Response · Authors · 2024-08-13
> > >
> > > Thanks for your feedback.
> > >
> > > There are significant differences between SC-GS's motivation and ours:
> > > - SC-GS is proposed to solve the novel view synthesis problem for dynamic scenes, which introduces sparse control points together with an MLP for modeling scene motion. The main motivation is that we can use a sparse set of base to represent motions within the dynamic scene. Therefore, SC-GS does not involve articulated information.
> > > - Our approach focuses on the task of extracting explicit skeletons from images. According to the methods you listed, this task is important and challenging. And the motivation of ours is that we  can extract explicit skeleton from the learned motions of superpooints.
> > >
> > > Compared with the listed methods, our method **does not require any priors**, such as templates, 3D datasets, etc. Specifically,
> > > 1. To recover the skeletal shape of an animal from a monocular video, CASA[1] first retrieves the most relevant articulated shape from a 3D character assets bank.
> > > 2. To reconstruct, rig, and animate the 3D objects from single images, Object Wake-Up[2] trained on the dataset with rigged 3D characters. The used “ModelsResource-RigNetv1” dataset only focuses on limited categories, e.g., chair, sofa, desk, and table.
> > > 3. While RigNet [3] aims to produce animation rigs from input character models, it also requires the dataset of 3D articulated characters to train the model.
> > >
> > > We are working to incorporate SC-GS into our work to provide further evidence.

---

> > > > ### Author Response · Authors · 2024-08-14
> > > > **Performance Improvement**
> > > >
> > > > After intensive work, we have partially succeeded in integrating SC-GS into our work. The performance of the proposed method is greatly improved. Please refer to the tables below:
> > > >
> > > > **PSNR**:
> > > >
> > > > |  scene       | average |  hellwarrior |  hook  |  jumpingjacks |  mutant |  standup  | trex    |
> > > > | ----             |  ---------- | --------------  | -------- | ------------------ | ---------- | -----------  | -------  |
> > > > | ours (old)   | 34.31    |  37.59         | 33.73  |  32.09            | 36.70    |  35.91      | 29.82 |
> > > > | ours (new) | 39.07    | 39.69          | 36.35  | 37.38            | 41.27    | 43.12       | 36.58 |
> > > > | SC-GS       | 43.04    | 42.93           | 39.87  | 41.13            | 45.19    | 47.89       | 41.24 |
> > > >
> > > > **SSIM**:
> > > >
> > > > |  scene       | average |  hellwarrior |  hook   |  jumpingjacks |  mutant |  standup  | trex     |
> > > > | ----             |  ---------- | --------------  | --------  | ------------------ | ---------- | -----------  | -------   |
> > > > | ours (old)   | 0.9751   | 0.9667       | 0.9742 |  0.9746           | 0.9837  | 0.9797   | 0.9714 |
> > > > | ours (new) |  0.9876  | 0.9767       | 0.9829 |  0.9885           | 0.9927  | 0.9934   | 0.9912 |
> > > > | SC-GS       | 0.9975   | 0.994          | 0.997    | 0.998             | 0.999    | 0.999     | 0.998   |
> > > >
> > > > Due to time constraints, we were unable to perfectly integrate SC-GS into our work, and there is still a lot of room for improvement. But the results in the above table are sufficient to provide evidence for the assert we discussed above.
> > > >
> > > > After the discussion period, we will continue working on improving the performance by incorporating SC-GS and update the
> > > > improved results in the final revised paper.
> > > >
> > > > Thank you for your time and feedback again.

---

> > > > > ### Comment · Reviewer_57cM · 2024-08-14
> > > > >
> > > > > Thanks for the author's response and the experimental validation done in a short period of time, which demonstrates the practical value and potential of the method. I also agree that optimizing the skeleton structure of the object is meaningful.
> > > > >
> > > > > I recommend that the author review the aforementioned work on video-based skeleton optimization and consider follow-up work, as it would be meaningful for assessing the overall contribution of the paper. Additionally, having reviewed other reviewers' feedback, I decided to raise my score. However, I believe that the method does not introduce new approaches or achieve state-of-the-art performance in either dynamic Gaussian reconstruction or skeleton optimization, so I lean towards Borderline Accept.

---

### Official Review · Reviewer_xZeD · 2024-07-13

**Soundness:** 3
**Presentation:** 4
**Contribution:** 4
**Rating:** 7
**Confidence:** 4

**Summary:**

The paper proposes a novel approach for reconstructing reposable dynamic 3D objects from RGB videos using Gaussian Splatting, without requiring any template as input.

To achieve this, the paper suggests grouping Gaussians around superpoints, which are intended to represent rigid parts of the scene. By optimizing and analyzing these superpoints, a full skeleton model of an articulated object in the input video can be built, refined, and used for reposing purposes.

**Details**

The approach consists of two main stages:

1. *Dynamic Stage*. After optimizing a canonical 3D Gaussian Splatting (3DGS) representation for a few iterations, a set of superpoints is initialized in the scene. A deformable field mapping each superpoint to a time-variant 6DoF transformation is optimized. These transformations are used to derive the motion of Gaussians by interpolating transformations with neighboring superpoints through linear blend skinning (LBS). The paper also proposes a gradient-based strategy to control (prune, merge, or densify) the number of superpoints in the scene. Toward the end of the dynamic stage, a skeleton structure with joints is enforced and discovered in the scene by analyzing the distance between and configuration of superpoints.

2. *Kinematic Stage*. After discovering the skeleton model of the scene, the number of Gaussians and superpoints is fixed and optimized along with a new MLP mapping skeleton joints to time-variant rotation matrices. These matrices are used to compute the motion of each Gaussian along the kinematic chains using LBS. After full optimization, the skeleton can be used for reposing and editing the reconstructed object.

The paper presents extensive experiments and demonstrates higher rendering performance and speed compared to concurrent reposable dynamic Radiance Field methods.

**Strengths:**

1. The paper is well-written and easy to follow.

2. The task addressed by the paper is challenging but crucial for many applications in both graphics and robotics. I appreciate the proposed strategy, which successfully retrieves kinematic chains from RGB videos.

3. The quantitative evaluation presented in the paper is convincing and clearly demonstrates the superiority of the approach over concurrent methods.

**Weaknesses:**

1. The paper may lack sufficient skeleton examples to effectively demonstrate that the proposed approach can recover meaningful structures from RGB videos. Indeed, the primary goal is to recover skeleton structures and enable reposing capabilities, but only a single skeleton example is provided (Figure 5). Including more qualitative examples would likely make the paper more convincing.

2. The paper does not provide details on the optimization time and required resources (e.g., VRAM) for the proposed approach. It appears that a large number of training iterations is needed; a comparison with previous state-of-the-art models would be valuable.

3. The limitations of the approach are interesting but are only discussed in the supplementary material, which is problematic in my opinion. These limitations are crucial for further research and should be included in the main text.

**Questions:**

1. What are the optimization time and required resources (e.g., VRAM) for the proposed approach? How does it compare to state-of-the-art methods?

2. Do the authors have insights on how the method would perform in the context of a monocular video with a moving camera, which is a more realistic setting than multi-view videos?

3. In Figure 5, the skeleton appears to be quite accurate, but some bones are located outside the geometry. Would it be possible to enforce the skeleton to be located “inside” the geometry, perhaps by applying a penalty that encourages superpoints to be close to the centroid of their associated Gaussians?

**Limitations:**

As I already mentioned it, the limitations are only discussed in the supplementary material, which is problematic in my opinion. I encourage the authors to try to move the limitations to the main paper in the final version.

---

> ### Author Rebuttal · Authors · 2024-08-06
>
> Thanks for your careful and valuable comments. Below are our responses to the specific points raised.
>
> - **Q1/W2**: Optimization time and required resources
>
>   **Answer**: Similar to 3D-GS, the optimization time and required resources are dependent on the number of Gaussians. For the D-NeRF dataset, the average optimization time is 6.3 hours, and the average peak GPU memory is 4.08 GB. Please refer to Tables 1 and 3 in the attached `PDF` for details.
>
> - **Q2**: Performance on monocular video with a moving camera
>
>   **Answer**: The camera setup in the D-NeRF dataset can be considered as using a camera to capture 360-degree surround images centered on the target object. Therefore, using a monocular video with a moving camera as input, our method can achieve good performance. However, using forward-facing monocular videos as input may result in reconstruction failures or inaccurate skeletons.
>
> - **Q3**: Enforce the skeleton to be located "inside" the geometry
>
>   **Answer**: We acknowledge the suggestion to add an extra regularizer term to encourage the skeleton to be "inside" the geometry. We will explore this idea in our future work.
>
> - **W1**: More qualitative examples
>
>   **Answer**: We have provided additional qualitative examples in the supplementary video material.
>
> - **W3**: Limitations in the supplementary material
>
>   **Answer**: Thanks for your suggestion. We will move it into the main text in the final version.

---

> > ### Comment · Reviewer_xZeD · 2024-08-11
> > **Thanks!**
> >
> > I would like to thank the authors for the efforts they made during the rebuttal as well as the answers to my questions.
> >
> > I think the paper tackles a very challenging task (recovering skeletons with a template-free optimization pipeline) and provides an interesting solution relying on 3DGS.
> >
> > Even though the output skeletons are still over-complicated for potential applications in animation, the proposed approach is certainly a step forward for the field.
> > For these reasons, I recommend acceptance and decide to maintain my initial rating.

---

### Author Rebuttal · Authors · 2024-08-07

We thank the reviewers for their positive and constructive feedbacks.

The attached `PDF` contains 3 tables and 2 figures.

---

### Decision · Program_Chairs · 2024-09-25

**Decision:**

Accept (poster)

**Comment:**

The paper is about reconstructing a 3D skeleton model from dynamic images with 3DGS and superpoints. The reviews were initially split, with key weaknesses pointed out. An active authors-reviewers discussion clarified all points and an acceptance consensus was reached.